# Simultaneous Determination and Pharmacokinetics Study of Three Triterpenes from *Sanguisorba officinalis* L. in Rats by UHPLC–MS/MS

**DOI:** 10.3390/molecules27175412

**Published:** 2022-08-24

**Authors:** Fanshu Wei, Chunjuan Yang, Lihong Wu, Jiahui Sun, Zhenyue Wang, Zhibin Wang

**Affiliations:** 1Key Laboratory of Basic and Application Research of Beiyao, Ministry of Education, Heilongjiang University of Chinese Medicine, Harbin 150040, China; 2Department of Pharmaceutical Analysis and Analytical Chemistry, College of Pharmacy, Harbin Medical University, Harbin 150086, China

**Keywords:** UHPLC-MS/MS, triterpenes, *Sanguisorba officinalis* L., pharmacokinetics, transformation

## Abstract

A selective and rapid ultra-high-performance liquid chromatography–tandem mass spectrometry (UHPLC–MS/MS) method was established and validated for the determination of ziyuglycoside I, 3*β*,19*α*-dihydroxyurs-12-en-28-oic-acid 28-*β*-d-glucopyranosyl ester, and pomolic acid in rats after the oral administration of ziyuglycoside I, 3*β*,19*α*-dihydroxyurs-12-en-28-oic-acid 28-*β*-d-glucopyranosyl ester, pomolic acid, and *Sanguisorba officinalis* L. extract. The separation was carried out on an ACQUITY UPLC^®^HSS T_3_ column (2.1 mm × 100 mm, 1.8 μm), using methanol and 5 mmol/L ammonium acetate water as the mobile phase. The three compounds were quantified using the multiple reaction monitoring mode with the electrospray ion source in both the positive and negative mode. Liquid-liquid extraction was applied to the plasma sample preparation. Bifendate was selected as the internal standard. The intra-day and inter-day precision and the accuracy of the method were all within receivable ranges. The lower limit of quantification of ziyuglycoside I, 3*β*,19*α*-dihydroxyurs-12-en-28-oic-acid 28-*β*-d-glucopyranosyl ester, and pomolic acid were 6.50, 5.75, and 2.63 ng/mL, respectively. The extraction recoveries of analytes in rat plasma ranged from 83 to 94%. The three components could be rapidly absorbed into the blood (*T*_max_, 1.4–1.6 h) both in the single-administration group or *S. officinalis* extract group, but the first peak of PA occurred at 0.5 h and the second peak at 4–5 h in the *S. officinalis* extract. Three compounds were eliminated relatively slowly (*t*_1/2_, 7.3–11 h). The research was to establish a rapid, sensible, and sensitive UHPLC–MS/MS method using the multi-ion mode for multi-channel simultaneous mensuration pharmacokinetics parameters of three compounds in rats after oral administration of *S. officinalis* extract. This study found, for the first time, differences in the pharmacokinetic parameters of the three compounds in the monomer compounds and *S. officinalis* extract administration, which preliminarily revealed the transformation and metabolism of the three compounds in vivo.

## 1. Introduction

Chinese herbal medicine, which can be used as both food and medicine, is widely applied to the prevention and cure of various diseases and clinical presentations, with its low toxicity and other unique advantages. The understanding of natural compounds, toxicity, and side effects, and the safe use of Chinese herbal medicine, can be an effective adjuvant treatment for disease [1,2]. In addition to the efficacy of the drug, understanding the safety of the drug and the pharmacokinetic profile of the bioactive compounds in the drug is crucial for determining the potential for successful treatment [1,2]. *Sanguisorba officinalis* L. (*S. officinalis*), belonging to the Rosaceae family and Sanguisorba genus, has detoxification, analgesic [3], and hemostasis [4] functions, which are noted in the Chinese Pharmacopoeia (2020 edition). It also possesses antioxidative [5], anti-inflammation [6], antiviral [7], antibacterial [8,9], and anti-tumor effects [10], as well as having anti-proliferative effects against breast cancer [11], liver cancer [12], lung cancer [13], and colorectal cancer [14]. The major active constituents of *S. officinalis* include triterpenes [15], tannins, and flavonoids [16]. Triterpenes are the primary hemostatic ingredients in *S. officinalis*. Many academicians have focused on pharmacological studies, including antioxidative, anti-inflammatory, and anti-cancer activities [17,18,19].

Ziyuglycoside I (ZGI), 3*β*,19*α*-dihydroxyurs-12-en-28-oic-acid 28-*β*-d-glucopyranosyl ester (DGE), and pomolic acid (PA) are active ingredients separated from *S.*
*officinalis*, all of them are triterpenes, and many scholars have carried out extensive research into their pharmacological properties [20]. ZGI is one of the main triterpenes in *S. officinalis*, and has been deemed to have significant anti-inflammatory and anti-rheumatic effects [21]. ZGI has a good anti-cancer effect in RB WERI-Rb-1 cells by activating p53, suggesting that ZGI may be a potential compound for chemotherapeutic agents for retinoblastoma gene-related cancer [22]. It was reported that DGE and PA have protective effects against liver injuries. It was also found that DGE and PA significantly inhibit nitric oxide production [23]. Moreover, PA activated P-gp/ABCB1 pathway, downmodulated MRP1/ABCC1 activities, and partially reverted epithelial-mesenchymal transition inducement in both PC3 and PC3R cell lines [24,25].

Great progress has been made in pharmacological research with regards to triterpenes, but the pharmacokinetic profile of triterpenes is not well-understood. In the past few years, several detection methods were developed for the determination of triterpenes. For instance, an LC-MS/MS method was created to estimate astragaloside IV using standard addition calibration [26]. An LC–MS/MS method was set up to determine the pharmacokinetics of six triterpenes in rats [27]. The HPLC–MS/MS method was used for the determination of ziyuglycoside I and its metabolite ziyuglycoside II in rat plasma [28]. However, there have been few reports on methods of simultaneous determination and pharmacokinetics of ZGI, DGE, and PA from *S. officinalis.*

This study has some innovative content in the optimization of mass spectrometry conditions, and a pharmacokinetic study of three triterpenes together in vivo following the oral administration of an extract of *S. officinalis* was reported for the first time. Furthermore, ZGI and DGE may be transformed into PA by a hydrolysis reaction under various conditions, such as enzymatic reaction and bacterial fermentation in vivo. Consequently, the objective of this research was to establish a rapid and sensitive UHPLC–MS/MS method for simultaneous measurement and investigate pharmacokinetics of three compounds in rats after oral administration of *S. officinalis* extract. This research aimed to lay a foundation for the clinical application of *S. officinalis.*

## 2. Results

### 2.1. Optimization of UHPLC–MS/MS Conditions

Both the negative and positive ion modes were checked by different mobile phases. Both *m/z* 784.5 for ZGI and *m/z* 652.5 for DGE with [M + NH_4_]^+^ can acquire strong and steady signals. To improve the sensitivity and enhance the response, the negative mode of [M − H]^−^ at *m/z* 471.2 was more appropriate for PA. To obtain a good resolution, an ACQUITY UPLC^®^HSS T_3_ (2.1 mm × 100 mm, 1.8 μm) column was adopted to separate the three analytes. In summary, two scanning time intervals with positive and negative modes were adopted: positive (ESI^+^: 0–5.0 min) and negative (ESI^−^: 5.0–6.5 min) ion modes. The main reason for segmental detection of the three compounds was to protect the capillary. Finally, the separating conditions were set with methanol (mobile phase A) and 5 mM ammonium acetate (mobile phase B) as the mobile phase at a flow rate of 0.3 mL/min.

The gradient elution program was as follows: 0–4.5 min, 70–90% B; 4.5–6.0 min, 90% B; 6.0–6.3 min, 90–70% B; 6.3–6.5 min, 70% B. The total running time was 6.5 min, and the injection volume was 10 μL.

### 2.2. Method Validation

#### 2.2.1. Selectivity

The selectivity was appraised by comparing the chromatograms of blank plasma, a plasma sample spiked with LLOQ analytes and IS, and a plasma sample from rats following single oral administration of monomers and *S. officinalis* extract, as listed in Figure 1. The results suggested that there was no interference of endogenous substances during the retention time of the IS and analyte.

#### 2.2.2. Linearity and Lower Limits of Quantification

The typical equations of the calibration curves were *Y* = 1.522*X* + 1.390 × 10^−1^ for ZGI, *Y* = 4.734*X* + 3.194 × 10^−2^ for DGE, *Y* = 0.685*X* + 6.882 × 10^−3^ for PA, where *X* means the plasma concentration of analytes and *Y* represents the peak area ratio of analytes to IS. As shown in Table 1, Figure 2, the calibration curves for ZGI, DGE, and PA had good linearity over the concentration ranges of 6.50–2600, 5.75–2300, and 2.63–1050 ng/mL, with all correlation coefficients *r* > 0.9960. The results showed that the compounds were within the linearity ranges. The LLOQ of ZGI, DGE, and PA were 6.50, 5.75, and 2.63 ng/mL, respectively.

#### 2.2.3. Precision and Accuracy

The precision and accuracies of the three analytes in rat plasma at four levels are listed in Table 2. In this experiment, the intra-day and inter-day precision (*RSD*) of the three analytes were not more than 13%, and the accuracies (*RE*) for the three analytes ranged from −0.71% to 4.7%. The precision and accuracy were in line with the relevant provisions of the biological sample analysis guidelines.

#### 2.2.4. Matrix Effect and Extraction Recovery

The extraction recoveries and matrix effects of the three components in rat plasma are listed in Table 3. The extraction recoveries of the three ingredients in rat plasma were 83–94% at three QC levels. The extraction recovery of IS was 94%. The matrix effects were between 102 and 109% for the three compounds in rat plasma. There were no significant matrix effects for the three analytes in rat plasma.

#### 2.2.5. Stability

The stability of rat plasma under different storage conditions was determined. The results (Table 4) showed the three analytes were stable in plasma after three freeze–thaw cycles, and then at room temperature for 4 h. The stability of the analytes at the post-preparative stage also showed that the sample did not degrade significantly when it remained at 4 °C for 12 h. All compounds maintained steady levels for 2 weeks at −20 °C.

#### 2.2.6. Carryover Effect

The blank rat plasma sample was immediately analyzed after the injection of the highest concentration calibration standard sample. The interference in the blank rat plasma sample was 6.0% at LLOQ (<20%) and 0.85% of the response for IS (<5%), which suggested that no obvious carryover effect was found under the described conditions.

### 2.3. Pharmacokinetic Studies

The UHPLC–MS/MS method was successfully used in the pharmacokinetic studies of the three analytes after single dose administration of ZGI, DGE, PA, and *S. officinalis* extract in rats (0.55 g/kg, equivalent to 0.03, 0.013, and 0.007 g/kg of ZGI, DGE, and PA, respectively). Based on the body surface area calculations of people and the animals and equivalent dose conversion calculations, the dosage of the rats was 0.55 g/kg. The mean plasma concentration–time curves of compounds are listed in Figure 3 and Figure 4. The half-time (*t*_1/2_), maximum plasma concentration (*C*_max_), time to reach *C*_max_ (*T*_max_), and area under concentration–time curve (AUC_0→t_ and AUC_0→__∞_) were calculated as shown in Table 5, Table 6 and Table 7.

Notably, as Table 5, Table 6 and Table 7 show, the pharmacokinetic process of three analytes differed between the monomer groups and S. *officinalis* extract. The *T*_max_ of ZGI after consuming the extract was similar to that of after consuming pure ZGI. In contrast, the *T*_max_ of DGE after consuming extract is higher than that of the pure DGE, suggesting that DGE in extract needs a longer time for absorption. The *T*_max_ values of ZGI, DGE, and PA were 1.4 ± 0.20, 1.6 ± 0.20, and 4.2 ± 0.41 h in the S. *officinalis* extract, and 1.4 ± 0.20, 0.92 ± 0.20, 0.92 ± 0.20 h for the ZGI, DGE, and PA groups, respectively. The absorbance velocity of the ZGI, DGE, and PA was rapid; they reached *C*_max_ between 0.92 and 2.00 h after single dose administration of ZGI, DGE, PA, and S. *officinalis* extract, but the first peak of PA appeared at 0.5 h and the second peak reached *C*_max_ at 4.2 h after administration of *S. officinalis* extract, compared with PA group. The *C*_max_ of ZGI, DGE, and PA in the *S. officinalis* extract was 1091 ± 1.1 × 10^2^, 359 ± 51, and 703 ± 96 ng/mL, respectively. The *C*_max_ of ZGI, DGE, and PA of the monomer groups was 678 ± 47, 166 ± 21, and 262 ± 31 ng/mL. Moreover, the AUC_0→t_ of the three compounds was 2877 ± 1.1 × 10^2^, 1200 ± 76, and 3315 ± 89 ng·h/mL in the *S. officinalis* extract, and 1716 ± 3.1 × 10^2^, 394 ± 33, and 1021 ± 55 ng·h/mL from the ZGI, DGE, and PA groups, respectively. The AUC_0→∞_ of the three compounds in the *S. officinalis* extract was 3098 ± 1.2 × 10^2^, 1314 ± 1.5 × 10^2^, and 4026 ± 2.1 × 10^2^ ng·h/mL, and the AUC_0→∞_ of the three compounds in the monomer groups was 2166 ± 3.3 × 10^2^, 551 ± 47, and 1183 ± 92 ng·h/mL. The increasing *C*_max_, AUC_0→t_, and AUC_0→__∞_ showed the better absorption of the three compounds in the *S. officinalis* extract. The mechanism of the difference in pharmacokinetics between the monomer groups and *S. officinalis* extract remains unclear. It may be inferred that some ingredients in the *S. officinalis* extract may enhance the absorption of the three analytes. Furthermore, the *t*_1/2_ values of ZGI, DGE, and PA were 9.3 ± 2.4, 7.3 ± 2.7, and 11 ± 1.4 h in the *S. officinalis* extract. The *t*_1/2_ values of ZGI, DGE, and PA were 11 ± 0.95, 11 ± 0.79, and 8.8 ± 1.0 h in the monomer groups. It was revealed that ZGI and DGE in the S. *officinalis* extract were more easily eliminated and metabolized than in the monomer groups. In addition, from these analysis results, we could conclude that PA has a different absorption compared with ZGI and DGE. The *t*_1/2_ of PA after consuming the extract is higher than that of the pure PA, suggesting that PA in extract needs a longer time for elimination. The *T*_max_ of PA was at 4.2 h, which was longer than the other two compounds. However, the content of PA in S. *officinalis* was much lower than that of ZGI, yet the AUC_0→t_ and AUC_0→∞_ of PA were greater than those of ZGI and DGE. The reason for these results was probably that ZGI and DGE metabolized PA in vivo.

## 3. Discussion

Few previous studies indicated that a method was developed to simultaneously determine ZGI, DGE, and PA from *S. officinalis* in the rat. Therefore, this research established a rapid, sensible, and sensitive UHPLC–MS/MS method for simultaneous determination and investigation of three compounds in rats after oral administration of *S. officinalis* extract. Meanwhile, the rats were given the three monomers; respectively, it was found that the three monomers have a transformational relationship and affect their pharmacokinetic parameters, to explore the process of the transformation of the three components in the rat.

For rapid, sensible, and simultaneous determination of three compounds, a UHPLC–MS/MS method was optimized. First, different proportions of mobile phase systems were tested, and ammonium acetate (5 mM) water was opted for, to enhance the peak strength of the three analytes. To realize a high extraction yield and weaken the matrix effect, LLE was chosen as the sample extraction method. As a result of the three compounds with similar polarities, ethyl acetate was the best option for extraction efficiency and repeatability. In addition, it is worth mentioning that to improve the sensitivity and accuracy, the mass spectrometer adopts multi-channel positive and negative ion mode to simultaneously detect a variety of compounds, which greatly optimizes the UHPLC–MS/MS method.

The elimination rate of the three compounds can be reflected from the *t*_1/2_. The *t*_1/2_ of both ZGI and DGE monomers were lower than those of *S. officinalis* extract. The reason may be that the presence of ZGI and DGE in the extract may be converted into PA in vivo, increasing the amount of PA and reducing the elimination rate of PA. In contrast, the conversion of ZGI and DGE in *S. officinalis* extracts reduces the amount in the body, so the elimination rate of these two compounds was improved. From the *T*_max_, the absorption rates of the three compounds can be reflected. It is worth noting that the DGE in *S. officinalis* extract was absorbed for a longer time, but the absorption amount was higher than that of the pure compound. The interactions between compounds of herbal extracts in vivo may affect the absorption of target compounds [29] The reason for the conversion of ZGI to DGE in the extract leads to its increased amount, prolonged absorption time, and increased absorption.

In addition, according to the data in Table 5, Table 6 and Table 7, in rats after the oral administration of ZGI, DGE, and PA, it is worth noting that pharmacokinetic parameters of three compounds may have some connection. As shown in Figure 5, ZGI may be converted into DGE and PA after oral administration of ZGI, and DGE may be transformed into PA after oral administration of DGE. On the contrary, PA cannot convert into ZGI and DGE after oral administration of PA. The hemiacetal hydroxyls of sugar molecules are active, so the linkage to triterpenes is unstable. This was previously demonstrated by the chemical and gut metabolic characterization of saponins [30]. C-3 and C-28 of pentacyclic triterpenoid saponins are prone to deglycosylation in vivo and in vitro [31]. The pentacyclic saponins can produce deglycosylation reactions in electric fields, as well as in intestinal flora in vitro [32,33,34]. On the other hand, a similar reaction also occurs in saponins in vivo [35]. However, the mechanism for this phenomenon needs further study. As shown in Figure 3 and Figure 4, the double peaks phenomenon was observed in the mean plasma concentration–time distribution curves of the three analytes during the elimination phase. The double peaks phenomenon of compounds may be due to the distribution of reabsorption and entero-hepatic circulation [36]. These results could be conducive to further exploring the mechanism of triterpenes, and provide effective pharmacokinetic information.

In this work, a rapid, sensible, and sensitive UHPLC–MS/MS method was established with the multi-ion mode, multi-channel simultaneous mensuration, and further applied to the pharmacokinetic studies of three compounds in rats after oral administration of *S. officinalis* extract. The pharmacokinetic and transformation characteristics of the three compounds in *S. officinalis* were initially revealed in vivo. However, the relevant mechanism of its specific metabolic behaviors needs further study.

## 4. Experiment

### 4.1. Chemicals and Reagents

The ZGI, DGE, and PA, with purity of more than 98%, were refined in our laboratory (identified by NMR and MS) [37]. Bifendate (Lot: 73536-69-3; purity > 98%, IS) was purchased from Chengdu Must Bio-Technology (China). Methanol (HPLC-grade) was obtained from J&K Medical (Beijing, China). Ammonium acetate was purchased from Kermel (Tianjin, China). Ultra-pure water was obtained using a Milli-Q water purification system (Millipore, Molsheim, France). All the other reagents, including ethyl acetate, ether, and dichloromethane, were of analytical grade. *Sanguisorba officinalis* L., which was purchased from the Anguo Traditional Chinese Medicine Market of Hebei, was appraised by Professor Zhenyue Wang of Heilongjiang University of Chinese Medicine.

### 4.2. Animals

The experimental protocol was permitted by the Animal Ethics Committee of Harbin Medical University (Approval Code: AECHMU20150021, Approval Date: 3 June 2015) and conformed to the principles for the Care and Use of Laboratory Animals. Twenty-four male Sprague-Dawley rats (weighting 200 ± 20 g), which were provided by the Laboratory Animal Center at Harbin Medical University (Harbin, China), were used for the pharmacokinetic study. The animals were maintained in a SPF-grade room on a 12 h light/dark cycle with a controlled temperature conditions (21 ± 2 °C) and relative humidity (50 ± 5%) before the experiment. During the adaptation process, all rats were provided with standard food and purified water each morning and evening. Each rat fasted for 12 h before giving the drug and was randomly divided into one of four groups (*n* = 6).

### 4.3. Preparation of S. officinalis Extract

After smashing the dried root of *S. officinalis*, it was weighed to 100 g. The *S. officinalis* was extracted by reflux with 2000 mL of 70% ethanol (1:10, *w/v*) solution two times at 85 °C, 1 h each time, and was then filtrated. The combined filtrate was evaporated into steam, and the residue was dissolved in water to obtain the concentration of the *S. officinalis* extract equivalent to 0.025 g/mL.

### 4.4. UHPLC–MS/MS Analytical Conditions

The analytes were evaluated on an Agilent series 1290 UHPLC instrument combined with an Agilent Technologies 6430 mass spectrometer. The eluant was surveyed by applying a triple quadrupole tandem mass spectrometer equipped with an ESI source and operated in positive (ESI^+^: 0.0–5.0 min) and negative (ESI^−^: 5.0–6.5 min) ion modes with MRM. The transitions were *m/z* 784.5→437.4 for ZGI, *m/z* 652.5→455.4 for DGE, *m/z* 471.2→453.2 for PA, and *m/z* 418.9→342.8 for bifendate (IS). The mobile phases were as follows: solvent A was 5 mM ammonium acetate water, and solvent B was methanol, which was delivered at a flow rate of 0.3 mL/min. The gradient elution program was as follows: 0–4.5 min, 70 to 90% B; 4.5–6.0 min, 90% B; 6.0–6.3 min, 90–70% B; 6.3–6.5 min, 70% B. The sample injection volume was 10 μL.

The conditions for MS analysis were as below: drying gas (N_2_) flow rate, 11 L/min; drying gas temperature, 300 °C; high purity nitrogen (N_2_) was atomized as the nebulizing gas; capillary voltage, 4000 V. The mass parameters for the three analytes and IS are listed in Table 8. The chemical structure and product ion scan spectra of ZGI, DGE, PA, and IS are presented in Figure 6.

### 4.5. Preparation of Calibration and Quality Control (QC) Samples

Standard stock solutions of the three components were obtained from resolving each compound in methanol to obtain an ideal concentration (0.52 mg/mL for ZGI, 0.23 mg/mL for DGE, 0.21 mg/mL for PA). Standard solutions were prepared by compatible dilutions of the stock solutions with methanol (6.5–2600 ng/mL for ZGI, 5.8–2300 ng/mL for DGE, 2.6–1050 ng/mL for PA). The IS solution (520 ng/mL) was prepared by diluting stock solution of methanol. Calibration standards were prepared by spiking each working stock solution at seven concentrations of 6.5, 13.0, 26.0, 130.0, 260.0, 520.0, and 2600 ng/mL for ZGI; 5.8, 11.5, 23.0, 115.0, 230.0, 460.0, and 2300 ng/mL for DGE; and 2.6, 5.3, 10.5, 52.5, 105.0, 210.0, and 1050 ng/mL for PA. Quality control (QC) samples were prepared at 13.0, 130.0, and 2080 ng/mL for ZGI, 11.5, 115.0, and 1840 ng/mL for DGE, and 5.3, 52.5, and 840 ng/mL for PA. All solutions were immediately stored at 4 °C.

### 4.6. Sample Preparation

The supernatant was separated by adding 10 μL IS (520 ng/mL) solution into 100 μL plasma sample, vortexing for 30 s, and mixing with 3 mL ethyl acetate by being vortex-mixed for 60 s. After centrifuging, the supernatant was dried by N_2_ blowing at 40 °C. After stirring for 4 min at 3800 rpm, the residue was reassembled with 100 μL methanol, then mixed by vortexing for 120 s and filtered by a 0.22 μm organic membrane. Finally, 10 μL sample solution was injected into the UHPLC–MS/MS system [38].

### 4.7. Method Validation

The method was appraised with the FDA guidelines, https://www.fda.gov/media/70858/download (accessed on 21 May 2018) [39].

#### 4.7.1. Selectivity

Selectivity evaluation was conducted by comparing chromatograms of blank plasma samples from six individual rats, corresponding blank plasma spiked with LLOQ of the three analytes and IS, and the plasma samples from the rats after oral administration of the three analytes and *S. officinalis* extract.

#### 4.7.2. Linearity and LLOQ

The calibration curves were constructed by plotting the peak area ratio versus the concentration of the three analytes and IS with a weighted (1/*x*^2^) least square linear regression, using standard plasma samples. The lower limit of quantification (LLOQ) was the lowest analytical concentration of the calibration curve, for which the signal-to-noise ratio was >10.

#### 4.7.3. Precision and Accuracy

The intra- and inter-day precision and accuracy were determined by testing the LLOQ sample and QC samples at three concentration levels of ZGI, DGE, and PA in six replicates for three days in a row. Precision and accuracy were shown by the coefficient of variation (RSD) and relative error (RE), respectively. The RSD values should not exceed 15% for the QC samples, except for the LLOQ, which should not exceed 20%. The RE values should be within ±15% of the nominal values for the QC samples, except for the LLOQ, which should be within ±20% of the nominal value.

#### 4.7.4. Matrix Effect and Extraction Recovery

The extraction recovery of analytes was determined by comparing the peak areas of the three analytes from the QC samples with those obtained from blank plasma samples with the three analytes spiked into the post-extraction supernatant at three QC levels in six replicates. The matrix effect was evaluated by comparing the peak areas of analytes spiked after plasma extraction into the post-extraction supernatants at three QC levels in six replicates. The extraction recovery and matrix effects of IS were also determined at one concentration. The RSD values should be within ±15% of the nominal values for the QC samples and IS, except for the LLOQ, which should be within ±20% of the nominal value.

#### 4.7.5. Stability

The stability of the three compounds in rat plasma, including freeze and thaw stability (three freeze-thaw cycles at −20 °C), long-term stability (storage for 2 weeks at −80 °C), room temperature stability (storage for 4 h at ambient temperature), and post-preparation stability (storage for 12 h after sample preparation at 4 °C), was tested at two QC levels with six replicates at each level. All stability testing QC samples were determined using the calibration curve of freshly prepared standard samples.

#### 4.7.6. Carryover Effect

The carryover effect was assessed by injecting a blank rat plasma sample after calibration of the standard sample at the upper limit of quantification. Carryover in a blank rat plasma sample should not be greater than 20% of response at the LLOQ for samples and 5% of response for IS.

### 4.8. Pharmacokinetic Study

A single dose of the *S. officinalis* extract (0.55 g/kg, group I), ZGI (0.03 g/kg, group II), DGE (0.007 g/kg, group III), and PA (0.013 g/kg, group IV) was administrated to the rats. The *S. officinalis* extract was dissolved in water. A single dose of 0.55 g/kg of *S. officinalis* extract was administrated to the rats. Blood was obtained from the retinal venous plexus at 0, 0.083, 0.5, 1.0, 1.5, 2.0, 2.5, 3.0, 4.0, 6.0, 8.0, 12.0, and 24.0 h after dosing. The plasma was immediately separated by centrifugation at 8000 rpm for 4 min at −4 °C.

The maximum concentration (*C*_max_) and the time to attain it (*T*_max_) were observed directly from the measured data. The elimination rate constant (*K*_e_) was calculated by linear regression of the terminal points in a semi-log plot of the plasma concentration against time. The area under plasma concentration–time curve (AUC_0→t_) to the last measurable plasma concentration (*C*_t_) was estimated using the linear trapezoidal rule.

### 4.9. Data Analysis

Calculate the amount of three compounds using linear regression from the standard curve. The pharmacokinetic parameters of the analytes were calculated with DAS 2.0 (Shanghai, China) [40]. All results were expressed as mean ± SD. The comparison of pharmacokinetic parameters was conducted using standard Student’s *t*-test. Differences between groups were assumed statistically significant for *p* values < 0.05.

## 5. Conclusions

This study demonstrated that a simple, rapid, and sensitive LC–MS/MS method was successfully developed for simultaneous quantification of the three representative components from ZGI, DGE, PA, and *S. officinalis* in rat plasma. This is the first report of a pharmacokinetic study of these three triterpenes together in vivo following the oral administration of ZGI, DGE, PA, and *S. officinalis* extract. This paper may be useful for further studies on the metabolism and absorption process of *S. officinalis* extract in vivo, and may also be beneficial for the application of this TCM in clinical therapy.

## Figures and Tables

**Figure 1 molecules-27-05412-f001:**
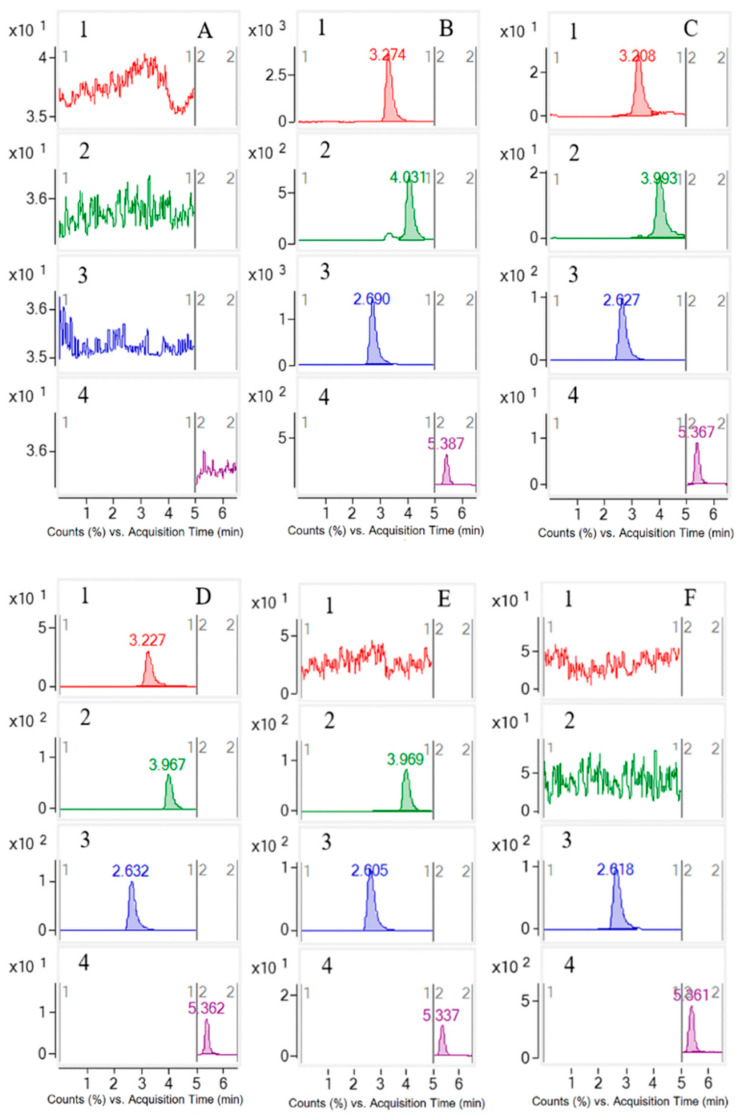
The chromatograms of ZGI (**1**), DGE (**2**), PA (**3**), and IS (**4**) in rat plasma samples. (**A**) Blank plasma; (**B**) plasma samples after oral administration of *S. officinalis* extract 1.5 h; (**C**) LLOQ sample (three analytes and IS in blank plasma); (**D**) plasma samples after oral administration of ZGI 1.5 h; (**E**) plasma samples after oral administration of DGE 1.0 h; (**F**) plasma samples after oral administration of PA 1.0 h.

**Figure 2 molecules-27-05412-f002:**
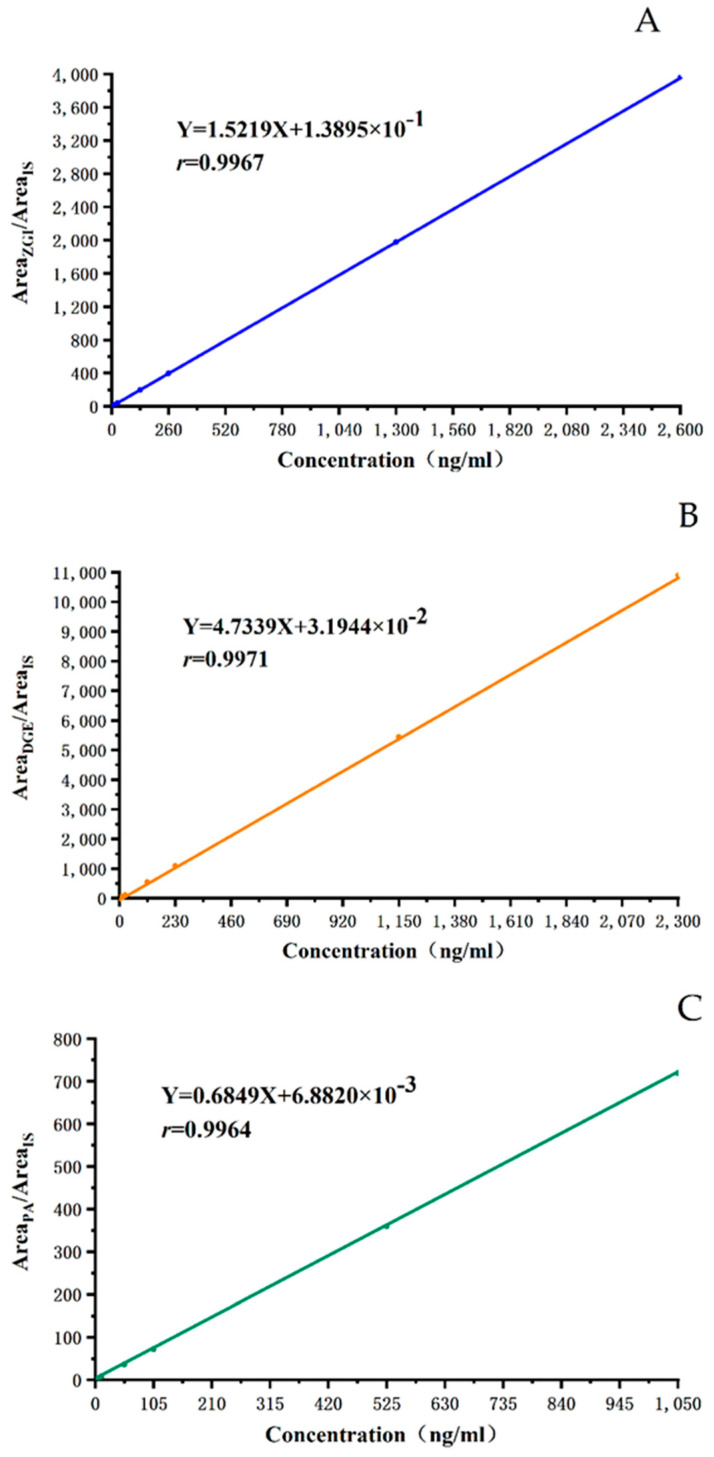
The calibration curves of ZGI, DGE, PA. (**A**) The calibration curves of ZGI (**B**) The calibration curves of DGE. (**C**) The calibration curves of PA.

**Figure 3 molecules-27-05412-f003:**
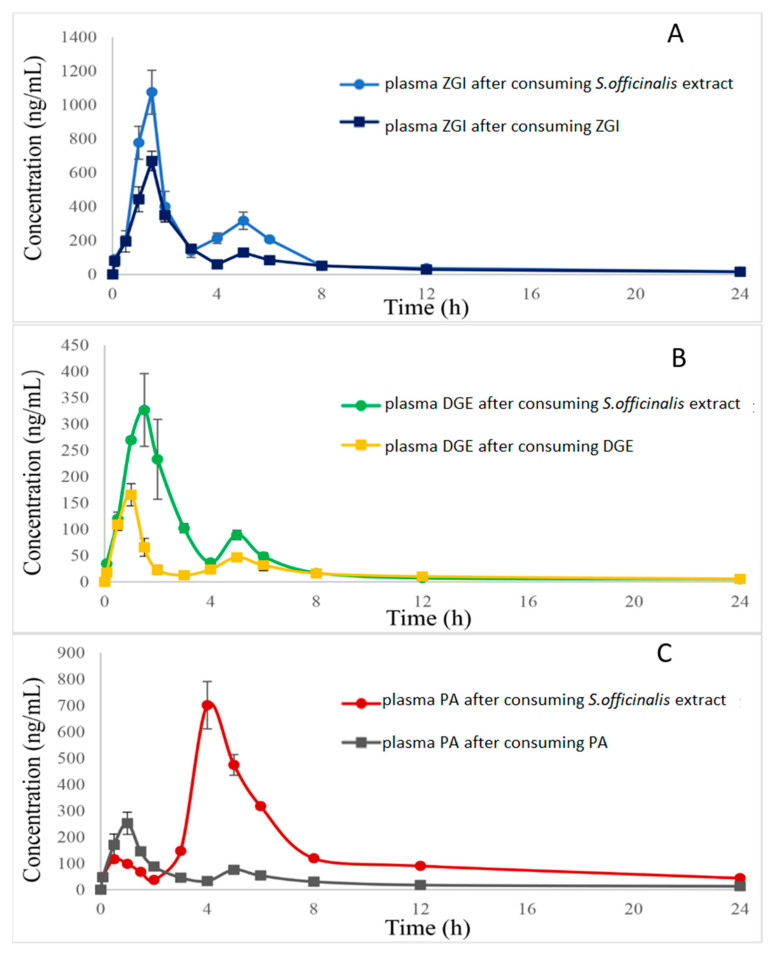
(**A**): Mean plasma concentration–time curve of ZGI in rats after oral administration of ZGI and *S. officinalis* extract; (**B**): Mean plasma concentration–time curve of DGE in rats after oral administration of DGE and *S. officinalis* extract; (**C**): Mean plasma concentration–time curve of PA in rats after oral administration of PA and *S. officinalis* extract.

**Figure 4 molecules-27-05412-f004:**
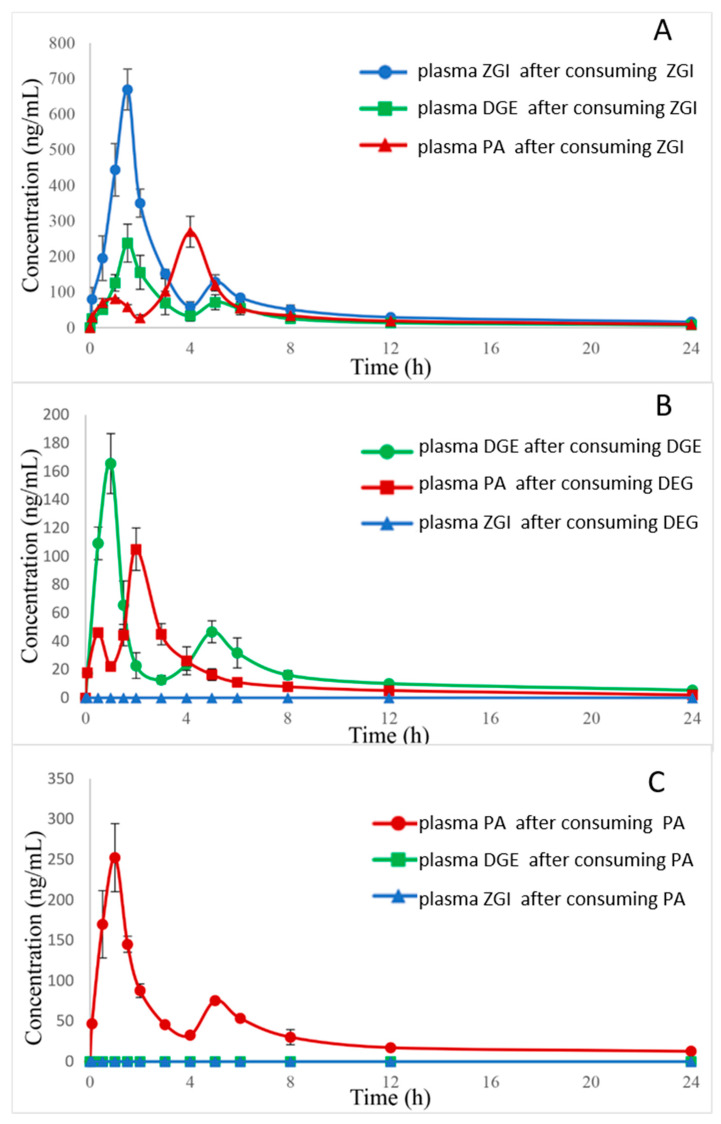
(**A**): Mean plasma concentration–time curve of the three compounds in rats after oral administration of ZGI; (**B**): Mean plasma concentration–time curve of the three compounds in rats after oral administration of DGE; (**C**): Mean plasma concentration–time curve of the three compounds in rats after oral administration of PA.

**Figure 5 molecules-27-05412-f005:**
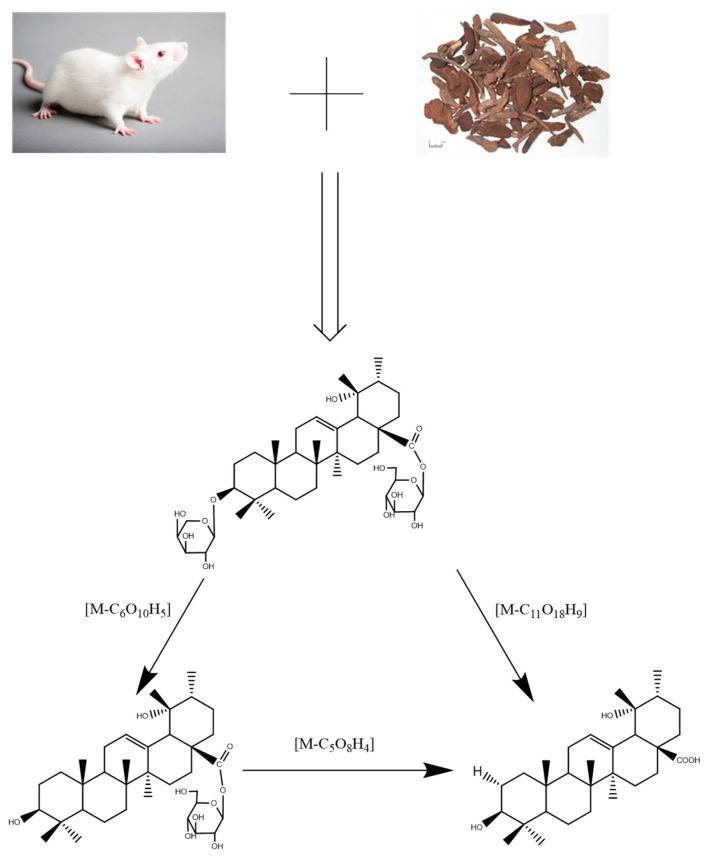
Proposed metabolic pathways of ZGI, DGE, and PA in rats.

**Figure 6 molecules-27-05412-f006:**
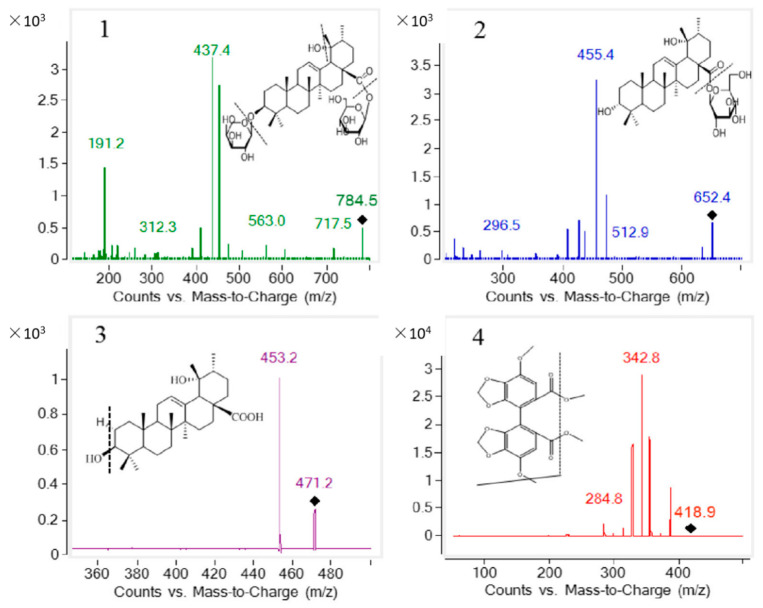
Product ion mass spectra of ZGI (**1**), DGE (**2**), PA (**3**), and bifendate (IS) (**4**).

**Table 1 molecules-27-05412-t001:** The regression equations, linear ranges, and LLOQ for the determination of analytes in rat plasma (*n* = 6).

Compounds	Regression Equation	*r*	Linear Range (ng/mL)	LLOQ (ng/mL)
ZGI	*Y* = 1.5219*X* + 1.3895 × 10^−1^	0.9967	6.50–2600	6.50
DGE	*Y* = 4.7339*X* + 3.1944 × 10^−2^	0.9971	5.75–2300	5.75
PA	*Y* = 0.6849*X* + 6.8820 × 10^−3^	0.9964	2.63–1050	2.63

**Table 2 molecules-27-05412-t002:** Intra-day and inter-day precision and accuracies for the determination of the three analytes in rat plasma (*n* = 6).

Compounds	Nominal Concentration (ng/mL)	Measured Concentration (ng/mL)	Accuracy (*RE*%)	Precision (*RSD*%)	
				Intra-Day	Inter-Day
ZGI	6.5	6.8 ± 0.84	4.7	12	13
	19.5	19 ± 1.2	2.4	8.7	11
	1300.0	1320 ± 10	1.7	6.4	13
	2080	2069 ± 116	−0.50	4.6	10
DGE	5.8	5.8 ± 0.60	0.89	10	12
	17.4	12 ± 1.0	1.8	9.0	8.0
	1150.0	1160 ± 5.7	1.5	5.0	3.3
	1840	1842 ± 80	0.15	4.1	5.5
PA	2.6	2.7 ± 0.32	1.8	12	12
	7.8	5.2 ± 0.38	−0.71	7.2	7.5
	525	537 ± 4.1	2.3	7.8	6.7
	840.0	845 ± 46	0.60	5.8	2.0

**Table 3 molecules-27-05412-t003:** Matrix effects and extraction recoveries for analytes and IS in rat plasma (*n* = 6).

Analytes	Concentration (ng/mL)	Extraction Recovery (%)	Matrix Effects (%)
		Mean ± SD (%)	Mean ± SD (%)
ZGI	19.5	87 ± 6.1	108 ± 3.7
	1300.0	91 ± 3.0	108 ± 3.4
	2080	94 ± 3.0	103 ± 7.2
DGE	17.4	87 ± 6.1	107 ± 4.9
	1150.0	92 ± 2.7	108 ± 3.7
	1840	94 ± 3.9	109 ± 16
PA	7.8	83 ± 6.9	107 ± 13
	525.0	88 ± 5.2	102 ± 11
	840.0	92 ± 7.8	107 ± 5.5
IS	520	94 ± 2.5	97 ± 5.2

**Table 4 molecules-27-05412-t004:** Stability of the three analytes in rat plasma under various conditions (*n* = 6).

Analytes	Spiked Concentration (ng/mL)	Stability (% *RE*)			
		Short-Term	Long-Term	Three Freeze -Thaw	Post -Preparative
ZGI	19.5	14	15	6.2	9.2
	2080	2.4	3.4	2.8	6.4
DGE	17.4	8.4	9.6	3.3	14
	1840	0.70	5.8	3.5	7.7
PA	7.8	−5.0	13	12	7.1
	840.0	3.1	7.6	5.2	13

**Table 5 molecules-27-05412-t005:** Mean plasma concentration–time curve of ZGI between those of the *S. officinalis* extract and the pure ZGI administration. (*n* = 6) (Compared with *S. officinalis* extract; ** *p* < 0.001).

ZGI	*C*_max_ (ng/mL)	*T*_max_ (h)	*t*_1/2_ (h)	AUC_0→t_ (ng·h/mL)	AUC_0→__∞_ (ng·h/mL)
*S. officinalis* extract	1091 ± 1.1 × 10^2^	1.4 ± 0.20	9.3 ± 2.4	2877 ± 1.1 × 10^2^	3098 ± 1.2 × 10^2^
ZGI	678 ± 47 **	1.4 ± 0.20	11 ± 1.0 **	1716 ± 3.1 × 10^2^ **	2166 ± 3.3 × 10^2^ **

**Table 6 molecules-27-05412-t006:** Mean plasma concentration–time curve of DGE between the *S. officinalis* extract, pure DGE, and pure ZGI (*n* = 6) (Compared with *S. officinalis* extract * *p* < 0.05; ** *p* < 0.001).

DGE	*C*_max_ (ng/mL)	*T*_max_ (h)	*t*_1/2_ (h)	AUC_0→t_ (ng·h/mL)	AUC_0→__∞_ (ng·h/mL)
*S. officinalis* extract	359 ± 51	1.6 ± 0.20	7.3 ± 2.7	1200 ± 76	1314 ± 1.5 × 10^2^
DGE	166 ± 21 **	0.92 ± 0.20 *	11 ± 0.79 **	394 ± 33 **	551 ± 47 **
ZGI	258 ± 33 **	1.6 ± 0.20	10 ± 1.7 *	728 ± 61 **	937 ± 58 **

**Table 7 molecules-27-05412-t007:** Mean plasma concentration–time curve of PA between the extract, pure PA, pure ZGI, and pure DGE. (*n* = 6) (Compared with *S. officinalis* extract * *p* < 0.05; ** *p* < 0.001).

PA	*C*_max_ (ng/mL)	*T*_max_ (h)	*t*_1/2_ (h)	AUC_0→t_ (ng·h/mL)	AUC_0→__∞_ (ng·h/mL)
*S. officinalis* extract	702 ± 96	4.2 ± 0.41	11 ± 1.4	3315 ± 89	4026 ± 2.1 × 10^2^
PA	262 ± 31 **	0.92 ± 0.20 **	8.8 ± 1.0 *	1021 ± 55 **	1183 ± 92 **
ZGI	271 ± 46 **	3.8 ± 0.41 **	9.2 ± 1.2 **	842 ± 51 **	1099 ± 1.4 × 10^2^ **
DGE	101 ± 17 **	2.2 ± 0.41 **	8.9 ± 0.62 *	268 ± 25 **	335 ± 20 **

**Table 8 molecules-27-05412-t008:** Mass spectrometric parameters of the three compounds and IS.

Compounds	Precursor Ion (*m/z*)	Product Ion (*m/z*)	Fragment (V)	Collision Energy (V)	Polarity
ZGI	784.5	437.4	150	10	positive
DGE	652.5	455.4	130	20	positive
PA	471.5	453.2	190	30	negative
IS	418.9	342.8	78	18	positive

## Data Availability

Not applicable.

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
