# Peer review of "Simultaneous Determination and Pharmacokinetics Study of Three Triterpenes from Sanguisorba officinalis L. in Rats by UHPLC–MS/MS"

_molecules, 2022, doi:10.3390/molecules27175412_

Round 1
Reviewer 1 Report
The paper describes a pharmacokinetic study that is novel and interesting reading for those working in the field of traditional chineese medicine.
There are several issues with the paper which needs to be corrected:
1. English language of the paper must be improved. For instance, the following parts are totally incomprehensible and/or not satisfactory:
-line 34: "...of various illnesses in clinical..."
-line 35: "The selection of natural, toxic, and side effects and safe use of Chinese herbal medicine is an effective method..."
-line 42: Just repetition of claims in the sentence before "..anti-inflammatory, antioxidant..."
-line 73: "sensible UHPLC-MS/MS method" should be "sensitive"
-line 74:"mensuration" a very uncommon word in analytical chemistry, perhaps "measurement" is better?
2. Reference style does not seem correct for Molecules, and all references are lacking the DOI.
3. All numbers must be checked in the text, tables and figures. The numbers are often with too many significant figures and must be rounded. Standard deviations (SD) not to be presented with more than 2 significant figures, and the mean rounded accordingly. If the authors do not know how to round their numbers correctly, this paper will appear to be of poor scientific quality and cannot be published.
Example in Table 2: 1160.7 ± 5.65 can be rounded to 1161 ± 6
Likewise the number 1842 ± 78.78 can be rounded to 1842 ± 80
4. Some errors detected:
-line 61: "For instance, a LC-MS method was created to estimate ziyuglycoside â… and ziyuglycoside â…¡ in rat plasma (27)" should be corrected to "For instance, a LC-MS/MS method was created to estimate astragaloside IV by using standard addition calibration (27)"
5. More info on blood collection is needed in line 349. Volume? Type of plasma and container for collection?
6. There are many collections of blood over 24h, perhaps as much as 3-5 mL blood in total, so how did the animals tolerate this? It is a rather large volume of blood taken from the 0.2 kg rats is it not? Please add a sentence on the welfare of the animals used in this study.
Author Response
Point 1: English language of the paper must be improved. For instance, the following parts are totally incomprehensible and/or not satisfactory:
-line 34: "...of various illnesses in clinical..."
-line 35: "The selection of natural, toxic, and side effects and safe use of Chinese herbal medicine is an effective method..."
-line 42: Just repetition of claims in the sentence before "..anti-inflammatory, antioxidant..."
-line 73: "sensible UHPLC-MS/MS method" should be "sensitive"
-line 74:"mensuration" a very uncommon word in analytical chemistry, perhaps "measurement" is better? "
Response 1: Thanks to the editor for the revision comments. The following locations have been revised accordingly following extensive English revisions
-line 34: "...of various illnesses in clinical..." change to’’ various diseases’’
-line 35: "The selection of natural, toxic, and side effects and safe use of Chinese herbal medicine is an effective method..." change to’’ The understanding of natural compounds, toxicity, and side effects, and the safe use of Chinese herbal medicine, can be an effective adjuvant treatment for disease.’’
-line 42: Just repetition of claims in the sentence before "..anti-inflammatory, antioxidant..." delete "..anti-inflammatory, antioxidant..."
-line 73: "sensible UHPLC-MS/MS method" change to "sensitive"
-line 74:"mensuration" change to "measurement"
We also did extensive article editing at MDPI, and the honorary certificate is attached ( English-Editing-Certificate-47845).
Point 2: Reference ,"Reference style does not seem correct for Molecules, and all references are lacking the DOI”.
Response 2: Thanks to the editor for the revision comments.
The doi number has been supplemented in the returned references.(line 401 to line 493)
Point 3: All numbers must be checked in the text, tables and figures. The numbers are often with too many significant figures and must be rounded. Standard deviations (SD) not to be presented with more than 2 significant figures, and the mean rounded accordingly. If the authors do not know how to round their numbers correctly, this paper will appear to be of poor scientific quality and cannot be published.
Example in Table 2: 1160.7 ± 5.65 can be rounded to 1161 ± 6
Likewise the number 1842 ± 78.78 can be rounded to 1842 ± 80
Response 3: Thanks to the editor for the revision comments.
We have rounded the results of the methodology and retained two significant figures to obtain more scientifically rigorous data (see details in 2.2. Method validation)
Point 4: General Information Section, "Some errors detected: -line 61: "For instance, a LC-MS method was created to estimate ziyuglycoside â… and ziyuglycoside â…¡ in rat plasma (27)" should be corrected to "For instance, a LC-MS/MS method was created to estimate astragaloside IV by using standard addition calibration [26]".
Response 4: Thanks to the editor for the revision comments.
We have revised and written according to the comments given (line 67 to line 68).
Point 5: General Information Section, “More info on blood collection is needed in line 349. Volume? Type of plasma and container for collection?”
Response 5: Thanks to the editor for the revision comments.
The blood samples (approximately 0.2 mL) were collected into heparinized centrifuge tubes (1.5ml). [Oral Pharmacokinetics of Enriched Secoisolariciresinol Diglucoside and Its Polymer in Rats. Yang X, Guo Y, Tse TJ, Purdy SK, Mustafa R, Shen J, Alcorn J, Reaney MJT.Nat Prod. 2021 Jun 25;84(6):1816-1822. doi: 10.1021/acs.jnatprod.1c00335. Epub 2021 May 27][Establishment of quantitative methodology for sophoridine analysis and determination of its pharmacokinetics and bioavailability in rat.Chen M, Jiang Q, Zhang M, Chen S, Lou J, Chen Y, Wang F, Wang R.Drug Dev Ind Pharm. 2021 May;47(5):741-747. doi: 10.1080/03639045.2021.1934862].
Point 6: General Information Section, “There are many collections of blood over 24h, perhaps as much as 3-5 mL blood in total, so how did the animals tolerate this? It is a rather large volume of blood taken from the 0.2 kg rats is it not? Please add a sentence on the welfare of the animals used in this study.”
Response 6: Thanks to the editor for the revision comments.
In designing the collection site, we did take full account of the health of rats (After feeding, the weight of rats all reached 220 g). After the experiment, the rats survived and were in good condition. We set up 12 time points within 24 hours with a total blood volume of 2.4 mL. Meanwhile, according to an authoritative reference, the well-accepted good practice recommended that no more than 20% of total blood volume [3.1 mL of 15.4 mL (~7% of body weight for 220 g rat)] should be collected via serial blood sampling within 24 h, and animal should receive fluid replacement (i.e. injection of sterile isotonic saline) equal to the volume collected to composite the blood loss in each sampling [Karl-Heinz Diehl et al., A Good Practice Guide to the Administration of Substances and Removal of Blood, Including Routes and Volumes. J. Appl. Toxicol. 21, 15-23 (2001)].
At the same time, we deliberately added a paragraph on animal welfare in Section 4.2
Reviewer 2 Report
The manuscript sounds interesting. However, it needs rigorous English editing. There are many unclear sentences that prevent readers from understanding your message.

Author Response
Point 1: The manuscript sounds interesting. However, it needs rigorous English editing. There are many unclear sentences that prevent readers from understanding your message.
Response 1: Thank you very much for your comments and opinions. We also did extensive article editing at MDPI, and the honorary certificate is attached ( English-Editing-Certificate-47845).
Reviewer 3 Report
| The research provide a simple, rapid, and sensitive LC-MS/MS method was developed for simultaneous quantification of the three components from ZGI, DGE, and S. officinalis in rat plasma. | 362 |
| Overall, the manuscript is technically sound and the research ideas appear justified. | |
Author Response
Point 1: The research provide a simple, rapid, and sensitive LC-MS/MS method was developed for simultaneous quantification of the three components from ZGI, DGE, and S. officinalis in rat plasma.
Overall, the manuscript is technically sound and the research ideas appear justified.
Response 1: Thank you very much for your comments and opinions.
Round 2
Reviewer 2 Report
The author did not edit the abstract. There are many sentences that are unclear.
line 25-26 This sentence is unclear esp. the first half. What does it mean.. three components could be rapidly absorbed in the blood between monomer groups and extract groups. Please edit the sentence.
line 27-28 Three compounds were eliminated 27 relatively slowly (t1/2, 7.32–11.24 h). Compare to what?
line 28-29 The conclusion is too general. We do not need your work to make such a conclusion. Please be more specific in summarizing the findings and significance of your work.
line 37-39 The edited sentence still does not make sense. The safety, toxicity, and natural compounds have nothing to do with the effectiveness of adjuvant treatment. Please edit it. Do you mean ... Besides the efficacy of the drug, understanding the safety of the drug and the pharmacokinetic profile of the bioactive compounds in the drug is crucial for determining the potential for successful treatment. ?
line 51 Ziyuglycoside I (ZGI), 3β,19α-dihydroxyurs-12-en-28-oic-acid 28-β-D-glucopyra-50 nosyl ester (DGE), and pomolic acid (PA) are triterpenes. Please edit it. You wrote that those are the active ingredients of triterpenes. Actually, they are triterpenes.
Line 56 You still did not edit. Please edit "human RB chemotherapy" to chemotherapeutic agents for retinoblastoma gene-related cancer
Line 58 Please spell out "NO" as nitric oxide
Line 58-59 The phrase "PA passed P-gp/ABCB1" are unclear. Please edit it to PA activated P-gp/ABCB1 pathway.
Line 59 Please spell out EMT to epithelial-mesenchymal transition
Line 62 Please edit the sentence "few studies have been conducted in the pharmacokinetic field" to "The pharmacokinetic profile of triterpenes is not well-understood. "
Line 65 Please edit to "an LC/MS-MS method"
Line 76 Please edit "such as enzymes and bacteria" to " such as enzymatic reaction and bacterial fermentation
Line 78 Please edit to "and investigate pharmacokinetics"
Line 91 Please specify which one is mobile phase A (is it methanol), and which one is B (is it 5 mM ammonium acetate).
Line 109 Please add figures to show calibration curves of ZGI, DGE, PA
Line 122 According to the data, the highest inter-day precision is 13%. Thus, please edit "were less than 13%" to "were not more than 13%". The 13% is not less than 13% but it was not more than 13%. Please report it accurately.
Figure 2-3 The label in the figure as ZGI of DGI, PA of PA, etc are very confusing. Please edit the labels in all figures as "plasma ZGI after consuming ZGI", "plasma ZGI after consuming the extract"
According to Table 6-7, why does giving pure ZGI and pure DGE raise the level of other compounds (DGE and PA) in the blood?
The presentation in Tables 5-8 makes it difficult to understand your result descriptions as "the pharmacokinetic process of three analytes differed between the monomer groups and S. Officinalis extract. " To address the difference, the tables should be revised in another way. Table 5 shall compare the pharmacokinetic parameters of ZGI between those of the S. Officinalis extract and the pure ZGI administration. Table 6 shall compare pharmacokinetic parameters of DGE between the extract, pure DGE, and pure ZGI. Table 7 shall compare the PK parameters of PA between the extract, pure PA, pure ZGI, and pure DGE. Also, the author must perform statistical analyses to show whether the differences in PK parameters between monomers and extract are statistically significant. You must show the p-value or * on the table as well.
Line 177-178 The description "The Tmax of ZGI and DGE was similar between the monomer groups and S. officinalis extract" is incorrect. According to the data presented, Tmax of DGE from extract is 1.6± 0.20 h, while Tmax of DGE from pure DGE is 0.92 ± 0.20 h. So the correct description should be "The Tmax of ZGI after consuming the extract was similar to that of after consuming pure ZGI. In contrast, the Tmax of DGE after consuming extract is higher than that of the pure DGE, suggesting that DGE in extract needs a longer time for absorption"
Line 194-198 "The mechanism of the difference in pharmacokinetics between the monomer 194 groups and S. officinalis extract remains unclear.....may affect the absorption of target 197 compounds [29]." should be in Discussion, not in the result section. Please edit it.
Line 195 The author stated that "It may be inferred that some ingredients 195 in the S. officinalis extract may enhance the absorption of the three analytes" Please add examples of such ingredients that can increase the absorption.
Line 202-203 The sentence "The reason for this may be that other ingredients of the S. officinalis extract 202 are conducive to the elimination of the two compounds." should be in the Discussion section not in the result. Please move it.
Line 203 Please add this statement. "The t1/2 of PA after consuming the extract is higher than that of the pure PA, suggesting that PA in extract needs a longer time for elimination.
Line 210-222 These paragraphs should be in the results sections, not the discussion.
The discussion section needs major revision.
At the beginning of the Discussion, please add a paragraph describing how your research fills the gap of knowledge that you raised in the introduction.
The second paragraph of the discussion, you should discuss how your methods of simultaneous determination of three compounds differ or resemble to previous studies.
The third paragraph of the discussion, please discuss the findings that the PK parameters of compounds in the extract are different from those of the pure compound. Also, please address the possible underlying reasons. Of note, you need to also discuss why DGE from extract needs a longer time for absorption but can absorb in a higher amount than that of the pure compound. And why PA from extract needs a longer time for elimination than that of the pure compound, while DGE and ZGI in the extract need shorter time for elimination than that of the pure compound.
The fourth paragraph of the discussion, you can describe the connection/ conversion among the three compounds and discuss the effect of those connection on PK parameters.
The last paragraph of the discussion, please add the strength and limitations of this research, and suggest future studies.
Section 4.9 line 367 Please add statistical analysis. Please describe how you calculate the amount of ZGI, DGE, PA, do you use linear regression from the standard curve? Also, please describe how you statistically compare the differences in PK parameters between those of extract and the pure compound
The conclusion needs revision. Please add a statement to summarize your findings for the difference in PK parameters between the extract and the pure compound.
